# Alleviating Heavy Metal Toxicity in Milk and Water through a Synergistic Approach of Absorption Technique and High Voltage Atmospheric Cold Plasma and Probable Rheological Changes

**DOI:** 10.3390/biom12070913

**Published:** 2022-06-29

**Authors:** Mohammad Ruzlan Habib, Shikhadri Mahanta, Yeasmin Nahar Jolly, Janie McClurkin Moore

**Affiliations:** 1Biological and Agricultural Engineering Department, Texas A&M University, College Station, TX 77843, USA; ruzlan277@tamu.edu (M.R.H.); shikhadri@tamu.edu (S.M.); 2Atmospheric and Environmental Chemistry Laboratory, Atomic Energy Centre, Dhaka 1000, Bangladesh; jolly_tipu@baec.gov.bd

**Keywords:** cold plasma, spectroscopy, milk, heavy metal removal, rheology

## Abstract

In this study, we combined atmospheric pressure cold plasma, a novel treatment technology, with an absorption technique with soybean husk to remove Pb and Cd from milk. Different combinations of treatment duration, voltage, and post treatment retention time were used to determine the effectiveness of cold plasma. Soybean husk was used for metal extraction, and it was observed that when the milk samples were plasma treated with a discharge voltage of 50 kV for 2 min and held for 24 h, the highest mean elimination of about 27.37% for Pb and 14.89% for Cd was obtained. Reactive oxygen and nitrogen species produced from plasma treatment were identified using Optical Emission Spectra analysis. A high voltage of 50 kV plasma for a 2 min duration could produce 500 ± 100 ppm of ozone concentration inside the treated package. The value of ΔE, which indicates overall color difference measurement, was significantly (*p* < 0.05) higher in all the treated samples than control samples. However, in the frequency range from 0.01 to 100 Hz, there was not much difference between the control and treated sample in the frequency sweep test. The identified functional groups at different wavenumbers (cm^−1^) in the treated samples were found to be similar compared to the control samples.

## 1. Introduction

Heavy metal contamination in foodstuffs is a great threat to human lives all over the world because of their ubiquity, natural bioaccumulation and biomagnification properties, and constant increase around the globe, which makes them a more serious concern [1,2]. The harmful effects of heavy metals are very different, ranging from nausea and vomiting after a short exposure all the way to death after a prolonged exposure. Chronic exposure to metals may result in a variety of carcinogenic, kidney, and cardiovascular diseases [3]. The main sources of heavy metals include industrial effluents, natural weathering of the earth crust, insecticides and pesticides applied to crops, mining, anthropogenic activities, sewage and domestic discharge, etc. [4,5,6]. Heavy metals tend to get into the body through various sources. These instances access the human body through food ingestion, inhalation, and dermal contact. Many strategies, such as bioremediation, adsorption techniques, cooking and processing practices, have been applied to remove heavy metals from wastewater and some foodstuffs worldwide [7,8,9]. However, the necessity for metal removal from foodstuffs still thrives due to a lack of effective removal techniques. With this thought that heavy metal pollution jeopardizes food safety issues, the removal of heavy metals thus gains greater importance to the researchers of the world. The WHO identified lead (Pb), cadmium (Cd), mercury (Hg) and arsenic (As) as the most concerning heavy metal for human health, as these are responsible for many acute and chronic diseases in human health [10,11,12,13]. In this study, only Pb and Cd were considered, and their mitigation procedure was proposed.

In 2018, the United States produced 34% of the world’s soybeans and led at the top [14], with 4.43 bushels produced. Because of its unique chemical composition, which includes approximately 8% hull, 2% hypocotyl axis, and 90% cotyledons, soybean is the most valuable and economic agricultural commodity [15]. Previously, several sections of the soybean plant were used to remove heavy metals from wastewater or other aqueous solutions using absorption or other metal removal techniques [16,17,18]. Soybean also demonstrated promising bio-sorbent efficacy in the removal of heavy metals from aqueous solutions [19]. As a result, soybean seed and hull could be used to remove heavy metals from milk.

Atmospheric pressure cold plasma (ACP) is a non-thermal treatment for foods that renders them safe to eat. Cold plasma has a wider range of applications that can help with a variety of food safety concerns. The success and versatility of cold plasma drives greater research to overcome more difficult challenges in human life. The most common reactive gas species produced from plasma using air are reactive oxygen and nitrogen species (RONS) such as hydrogen peroxide (H_2_O_2_), ozone (O_3_), superoxide anion (O_2_^*−^), hydroperoxyl (HO_2_^*−^), alkoxyl (RO^*^), peroxyl (ROO^*^), singlet oxygen (O), hydroxyl radical (^*^OH), carbonate anion radical (CO_3_^*−^), nitric oxide (NO^*^), nitrogen dioxide radical (^*^NO_2_), peroxy-nitrite (ONOO−), peroxy-nitrous acid (OONOH), and alkyl-peroxy-nitrite (ROONO) [20,21]. The wide applications of cold plasma or plasma-related mechanisms includes food safety by inactivating pathogens, improvement of food quality, contribution to seed germination and agriculture, packaging industry use, degradation of pesticide residue, medical treatment use and many more [22,23,24]. Along with the emerging implementation of plasma in food processing, such as hydrogenation of vegetable oils to yield trans-free edible oils, inactivation of anti-nutritional factors, and control of food allergens [25,26], new advances in ACP for food safety concerns are being experimented with. Similarly, the elimination of metals can be rendered and accelerated by ACP treatment. This was the focus of the current investigation, which makes it unique. The overall concept includes the interaction of plasma with the metal bonding of the milk and release the ions to be available for bio-sorbents absorption. Hence, the absorption technique can be coupled with the plasma treatment method for the separation process of metal from milk.

The objective of the study is to investigate the application of the atmospheric pressure cold plasma technique coupled with the absorption technique in reducing Pb and Cd toxicity from milk and water. The study will also focus on the rheological changes, i.e., the viscosity parameter of the milk after ACP treatment. Since there have been no previous studies that utilized ACP in heavy metal removal, the study provides novelty in investigation in the field of metal removal techniques.

## 2. Materials and Methods

### 2.1. Sample Preparation

The sampling matrix of the atmospheric pressure cold plasma treatment is shown in Table 1. In the table, the samples are named as a-b-c, where a = voltage applied (kV), b = treatment duration (min), and c = storage time/retention period (hours). For example, a 20-1-0 sample name means that the sample was treated at 20 kV for 1 min and stored for 0 h prior to analysis. For absorption and ACP treatment and analysis, fresh pasteurized milk was purchased from a local grocery store. For heavy metal infusion with the milk and water, 99.99% pure Pb and Cd in the form of Pb(NO_3_)_2_ and CdSO_4_ (Sigma-Aldrich, St. Louis, MO, USA) were purchased. For treatment and further analysis, 200 ppb of each of Pb and Cd were infused with milk and water. Both the samples (milk and water), 50 mL each, were placed in a Petri dish and soaked with 1 g soybean husk for 30 min to activate the absorption technique. A 0.1 mm-thick low-density polyethylene (LDPE) (221 mm × 175 mm) bag was used as package for the sample [27]. The bag was filled with modified atmospheric packaging gas (composition: 5.05% N_2_, 29.92% CO_2_, 65.03% O_2_) from Airgas Distributions (College Station, TX, USA), maintaining a bag height of 43 mm ± 5 mm to make it ready for ACP treatment.

### 2.2. Cold Plasma Treatment

An AC dielectric test set (Input: 120 VAC, 60 Hz, 1 Phase, 30 A; Output: 0–120 VAC, 45 A) (Phenix Technologies, Accident, MD, USA) was used for plasma generation, and it was coupled with a high voltage (HV) capacity of 0–120 kV range transformer (measurement: 120 kV/62.5 mA). The transformer was connected to the HV electrode (Figure 1) via a properly insulated flex wire, with grounded and parallel electrodes. Two dielectric barrier plates of 9.3 mm thickness were placed between the electrodes to keep the sampling bags at the proper height. The treatment was carried out at 20 kV and 50 kV (±0.8%) high voltages for 1 and 2 min application durations followed by instant digestion (within 0 to 30 min) and digestion after immediate storage of 24 h (Table 1) for metal analysis. For stored samples, the conditions were 4 °C and 55% RH to let the transitional species best interact with the sample.

### 2.3. pH and Ozone Concentration Test

To measure the ozone concentration produced in the treated sample packages, a toxic gas detector pump (Model: 8014-400B, Matheson Tri-Gas Inc., Montgomeryville, PA, USA) was used along with ozone measuring tubes (Range: 0 to 1000 ppm). The tubes are inserted into the pump, and the sample was inserted directly from the package to measure the ozone concentration (ppm). The pH was measured using standard testing strips (Fisherbrand, Thermofisher Scientific, Waltham, MA, USA), which were dipped into the sample for 10 s each. Three replicas of each treatment combination were conducted.

### 2.4. Optical Emission Spectroscopy (OES) Analysis

A concave grating BLACK-Comet Spectrometer (StellarNet Inc., Tampa, FL, USA) paired with an F600-UV-SR fiber optic cable connected to SpectraWiz software (version 5.33; StellarNet) on a computer was used to quantify a few of the created plasma gas species inside the package. The fiber optic cable had a 0.25 mm aperture and was directed to the package with a 7 mm gap between it and the electrodes, as well as a 19 mm-thick transparent plastic barrier to prevent electrical interference. The spectrometer wavelength ranged from 220 nm to 850 nm, with a detection integration limit of 1 millisecond to 8 min. The detection was performed at a 5000 ms integration time for the analysis, and the peaks were captured at different wavelengths and averaged over three samples. To reduce background noise, a pre-treatment background spectrum was determined and standardized. The spectra were evaluated and compared using the National Institute of Standards and Technology’s (NIST) atomic spectra database and published research papers [28,29].

### 2.5. Colorimeter Test

Before the ACP treatment of milk samples for heavy metal absorption, soybean husks were added to the milk as an absorbent. The milk samples were filtered and subjected to a colorimetry test to see if the color of the samples had changed as a result of the ACP treatment and the usage of soybean husks. The Hunter Lab’s Colorimeter was used to determine the color of the milk samples. During the colorimetry test, the samples were kept at room temperature. Samples were tested after the equipment was calibrated using black, white, and green tiles. A 1 cm-tall milk sample was put into a 5.5 × 2.5-inch glass plate and placed on the colorimeter’s aperture. The glass plate was then covered to prevent light from reflecting through the sample and potentially interfering with the results. Each sample was taken in triplicate, and the average of the measurements was calculated.

### 2.6. Rheological Analysis (Frequency Sweep)

A Rheometer (HAAKE RheoStress 6000, Thermo Scientific, Waltham, MA, USA) was used to conduct a dynamic oscillatory frequency sweep analysis in the linear viscoelastic region at frequencies of 0.01 Hz, 0.1 Hz, 1.0 Hz, 10.0 Hz, and 100.0 Hz to determine possible changes in the milk rheological parameter after ACP treatment and separation from the absorbent. To further understand the viscosity changes, the phase angle (tan) for the treated and control milk samples was calculated using the ratio of loss modulus (G”) and storage modulus (G′) [30,31]. The Rheometer was operated by software (Thermo Scientific’s HAAKE RheoWin version 3) and the outputs were evaluated using a data manager. Then, 1 cc of milk samples were distributed on the rheometer’s plate, and a cone (diameter 35 mm, angle 4°; Kegel mit: D = 35 mm, 4 GRD Winkel) was rotated (temperature controlled: 23 °C, gap 0.052 mm). All of the figures in the analysis are the average of two samples.

### 2.7. Sample Digestion and Metal Analysis

Following the procedure, liquid milk was removed from the samples and dried in a 95 °C oven for 24 h. A 0.3 g dried milk mass was then placed in a digestive tube. For water samples, an aliquot of 2 mL of water was poured into a digestion tube. The samples digestion tubes were placed in a hot block digester (HACH company, Loveland, CO, USA) and pre-digested with 2 mL of 3% (*v*/*v*) nitric acid (Certified ACS Plus) overnight, then heated at 95 °C for 4 h according to EPA method 3050b [32]. After being cooled down to room temperature, 2 mL of 30% (*w*/*v*) H2O2 was added to the samples and reheated in the hot block at 95 °C for an additional 2 h until fully digested [33]. After samples were diluted to a final concentration of 1% nitric acid, the Pb and Cd contents were determined using an inductively coupled plasma-mass spectrometry ICP-MS (Agilent 7500i, Agilent Technologies Co. Ltd., Colorado Springs, CO, USA).

### 2.8. Fourier-Transform Infrared (FTIR) Analysis

To see if there were any changes in the milk content after ACP treatment, a Nicolet iS50 FTIR spectrometer (Thermo Fisher Scientific, Madison, WI, USA) with an Attenuated Total Reflectance (ATR) was used. Prior to the sample examination, water was used for background analysis. Milk sample spectra were acquired using an XT-KBr beam splitter and DTGS ATR detector in the region of 4000–400 cm^−1^ wavenumbers, averaged from 32 scans with a resolution of 4 cm^−1^. The light source’s optical velocity for acquiring the spectra was 0.4747 with an aperture of 100 for sensitive detection and energy saturation prevention. The OMNICTM spectra program (Thermo Fisher Scientific, Madison, WI, USA), version 9.9.549 (2018), was used to gather and evaluate the spectrum.

### 2.9. Data Analysis

The IBM SPSS statistical program (version 21) and Microsoft Excel were used to analyze the data. All of the samples were tested for significance at the 0.05 level. The SPSS software was used to perform the non-parametric Kruskal–Wallis test for independent samples and compare the metal removal (%) between the voltage groups.

## 3. Results and Discussion

This section may be divided by subheadings. It should provide a concise and precise description of the experimental results, their interpretation, as well as the experimental conclusions that can be drawn.

### 3.1. Optical Emission Spectroscopy (OES) Analysis

As the feeding gas was a mixture of O_2_, CO_2_, and N_2_, major excited species generated from the plasma treatment were composed of reactive oxygen and nitrogen species (RONS) as compared with NIST. The OES of atmospheric pressure cold plasma and the upper and lower-level orbital configuration terms of the detected excited species are denoted in Figure 2. At 431.71 nm, a dominant oxygen species was observed as singlet oxygen (2S^2^2P^2^(3P)3S–2S^2^2P^2^(3P)3P). Ozone peak was observed at 373.48 nm and OH was at around 300 nm [34]. The peak for ozone looked wider, which may include similar excited species in nearby lines. The plasma chemistry varies depending on the feeding gas composition. The present observation showed more identifiable reactive oxygen species than nitrogen species. A carbon-based transient species line was observed at 476.67 nm, exhibiting the electron transition at the 4th orbital for an upper-level orbital configuration. Being an intermediate coupling, C(I) may have a potential role in transitioning -COOH and -RCOOH species [35]. Due to the huge number of continuous reactions occurring in the plasma chamber during the treatment, the diversity of species is also high, and the stability of the species is low. Some convoluted oxygen species were also observed at 700–750 nm [34]. Overall, a variety of RONS were generated in the presented plasma set up.

### 3.2. Ozone and pH Assay

After the samples were treated with ACP, ozone measurement is imperative to determine the efficacy of plasma application with the appropriate carrier gas. For a constant ACP duration, it is expected that higher voltage generates more ozone inside a package of oxygen (O_2_) contained gas. Since modified atmospheric packaging gas was used with 65% O_2_ while generating plasma, the above statement is observed to be true for the current study. For the same duration, a high voltage of 50 kV produced 25 to 32 times more ozone concentration than a low voltage of 20 kV (Table 2). Both Ozone and pH play a pivotal role for microbial decontamination from food substances when ACP is applied. Solubility of ozone in water is dependent on the temperature and pH [36], while the ozone solubility in milk is still unknown. However, if ozone is soluble in protein-based liquid foods, it can oxidize the polypeptide backbone of proteins, causing the disruption of protein–protein interaction [37]. The pH of both milk and water samples was slightly reduced upon ACP treatment, supporting some previous studies for DBD type ACP treatment on milk [38]. The increased treatment duration also decreased the pH of both samples for both 20 kV and 50 kV (Table 2). The decreased pH value results from the partially electrolyzed water molecules and water vapors from milk [39]. Additionally, the RONS species generated from the plasma may have been attributed to a pH decrease through transition-state reactions. Longer treatment of ACP can reduce the pH to 3.8 in water [40], favoring the heavy metal release from water. This mechanism can also be applied to milk, even though lower pH can alter the casein micelles of milk and the external surface layer [41].

### 3.3. Heavy Metal Concentration

After the samples were treated with ACP and the absorption technique, the samples were analyzed for metal concentration (Figure 3). For milk, the highest mean removal was found 27.37% for Pb and 14.89% for Cd when the sample was treated with 50 kV for 2 min and stored for 24 h. A negligible amount of metal removal was found when the milk samples were treated with 20 kV and there was no storage time. However, the higher voltage level and longer storage period were positively attributed to heavy metal removal from milk. The storage period did not positively impact the Pb reduction in milk for lower voltages, while for all cases, the storage period impacted the Cd removal. Hence, the optimal parameters for Pb and Cd removal from milk are treating the samples with plasma at 50 kV for 1 min and a 24 h retention period. It is known that heavy metals bind with food proteins as metallothionines and glutathione [42]. The removal (%) of Cd from milk was lower than Pb, probably due to the stronger chemical bonding of Cd to the metallothionines of the milk. It is also important to note that Pb and Cd were introduced to milk to identify the significant differences in concentration after plasma application. This induction may not have produced appropriate chemical bonding as the preoccupied metals, making removal more complex. To distinguish the rigidity and complexity of metal removal from milk, water was also considered for this research work. It was observed that both Pb and Cd can be reduced from water to a great extent, with79.85% and 74.06% removal, respectively, when water was treated with 50 kV ACP for 2 min and stored for 24 h to let the absorption work more. However, the storage period impacted the Cd removal for all voltage and time combinations, while Pb was impacted only for high voltages in milk (Figure 3). Furthermore, voltage differences had little effect on metal removal from water because a low voltage of 20 kV ACP treatment could achieve 58.47% Pb removal with no retention time and 69.58% Pb removal with 24 h of retention time.

The chemical structure of the metals with the milk protein is very complex (Figure 4). The proposed structure shows the binding of Cd with cysteinyl sulfur as a three-metal cluster. Each Cd^2+^ can be tetrahedrally combined with four cysteines to form a strong metallothionein bond. Low molecular peptides originating from cysteine may have complex bonding with Pb, causing the difficulty of removal from milk [43]. Simultaneously, casein and serum proteins are rich in milk protein, which may have a strong binding affinity with Pb^2+^, causing the removal to be inconvenient. Although Pb^2+^ has a higher affinity for TRI peptides than Cd^2+^, as observed in a study [44], the comparative bonding strength of Pb and Cd in milk proteins has yet to be determined.

The predicted mechanism for the removal process is that when the ACP was applied to milk, the metals were loosened from the metallothionein and the absorbent (soybean husk) absorbed some metals from the milk. Since the overall reaction is not long and the plasma produced RONS are transient, the interaction of the plasma with milk is expected not to be protein damaging or protein corrupting. It was previously studied that ROS can cleave the peptide bonds [46], which can support the loosening of metal bindings and, at the same time, can cause damage to milk proteins. Additionally, ACP treatment produces the most reactive oxygen species hydroxyl radical (^*^OH), which can attack the cellular molecules of RNA, protein, and DNA, causing destructive effects. This may also result in a reduction in the valence state of the metals in association with the Fenton reaction. However, since the current study did not observe high pH differences after ACP treatment, the above mechanism may not be applicable in this regard. Contrarily, ozone generation was distinguishably higher after the ACP treatment, especially for the 50 kV treatment, and may have impacted the Fenton reaction. Nonetheless, a deeper understanding of the appropriate mechanism is expected for future studies. Lead and Cadmium bonded with Thiols, a protein ligand, causes toxicity in some proteins [47], so future research may focus on thiol breakdown to eliminate Cd from them.

Non-parametric Kruskal–Wallis test was conducted to compare the mean removals of heavy metals from milk and water (Figure 5). The distribution of metal removal across the 50 kV voltage is significantly different (*p* < 0.05). This is also similar to the 20 kV treated samples. Mean Cd removal was higher in milk and water, while mean Pb removal was lower. At a low voltage of 20 kV, the metal’s removal was negligible.

### 3.4. Changes in Color of the Milk Samples

The color measured by the Hunter Lab colorimeter is given in a L^*^, a^*^, and b^*^ scale. Here, L^*^ gives information about brightness/luminance; values range from 0–100, where 100 is the measurement of the brightest and 0 is the darkest; a^*^ represents redness with values from −60 to +60; b^*^ value represents yellowness, measuring blue to yellow color on the scale of −60 to +60. Together, the L^*^, a^*^, and b^*^ values can represent an objective measurement of the color of the milk samples [48]. The differences in the colors of the different samples can be evaluated by calculating a ΔE component using the L^*^, a^*^, and b^*^ values [49] and are distinguished as non-noticeable (0–0.5), slightly noticeable (0.5–1.5), noticeable (1.5–3), visible changes (3–6), and great change (6–12) [50].

In a previous study, the colorimetric reading of raw whole milk was observed as L^*^: 88, a^*^: −0.88, and b^*^: 4.42 [51], where the measured control sample was different in L^*^, a^*^, and b^*^ values, supporting the current study (Table 3). From the samples measured, we can observe that the L^*^, a^*^, and b^*^ values of the treated samples are lower than the L^*^, a^*^, and b^*^ values of the control samples, demonstrating the slight deviation in color. The value of ΔE obtained was significantly (*p* < 0.05) higher in all the treated samples. The highest ΔE of 6.70 was measured when the sample was treated at 20 kV ACP for 2 min duration with 24 h of storage time, followed by ΔE of 6.61 when the sample was treated at 50 kV ACP for 2 min duration with 24 h of storage time. Earlier studies have shown that plasma treatment does not greatly affect the change in color in milk samples. Milk treated with 9 KV ACP for 20 min obtained the highest ΔE value of 0.52 only [49]. A similar ΔE score was obtained when milk was treated with plasma for 5 and 10 min, which resulted due to long plasma exposure to milk [52]. Nonetheless, relatively higher values of ΔE were found in several milk samples treated with plasma at 40V, 50V, 60V, 70V, and 80V for 120 s [38]. The ΔE values of the treated samples were higher, indicating that changes in color are visible. However, it is to be noted that the current study utilized a synergistic approach of absorption technique by soybean husk and ACP treatment, thus the color changes may have resulted from soybean husk. The presence of soybean husks in the milk samples during the plasma treatment and its prolonged immersion can affect the changes in color of the milk. Additionally, a decrease in the whiteness of the samples indicates a decrease in the opacity of milk as treated [53].

### 3.5. Rheological Properties

A frequency sweep (FS) test was conducted for 50 kV ACP-treated and absorbent-induced milk samples and the control milk sample to see the differences in rheological parameters of milk (Figure 6). Since the optimal voltage for efficient heavy metal removal was found at 50 kV, the FS test was conducted for the 50 kV only. The viscoelastic behavior of milk as a function of time can be measured using this FS test. Additionally, most of the liquid food processing requires the evaluation of quality control, structural change, and operational feasibility in the finished products. Hence, the FS test can provide the viscous behavior of treated milk, which in turn provides the quality of it for further processing. There has not been much dissimilarity observed between the control and treated samples in the frequency range from 0.01 to 100 Hz. In the frequency range, divided into four regions as terminal, rubbery plateau, transition, and glassy region, there was slight deformation noticed in the rubbery plateau region and glassy region for the storage modulus (G′). This probably impacted the absorbent induction to the milk sample. Added to this, loss modulus (G″) was also slightly lower at the end of the transition region for the treated sample. It is to be noted that the G′ and G″ measure the elastic and the viscous components of the sample, respectively. The viscous modulus was dominant over the elastic modulus until 5 Hz, and hence, the shifting properties are dependent on frequency for both samples. However, it was also observed that the complex viscosity (η^*^) pattern was similar at different frequencies for both the control and the treated samples, except for a smooth shift from 10 Hz to 20 Hz for the treated samples. Therefore, it can be elucidated that the treated samples showed minimal quality change in the milk in structure and viscosity.

Previously, a study reported a reduction in viscosity of about 6% when direct DBD plasma was applied to milk [54]. A complex output on viscosity differences was followed when different time and voltage combinations were applied to milk, as there was a decrease in viscosity for 15 to 90 s treatment time and an increase in viscosity for up to 120 s treatment time regardless of the voltage [38]. Thus, it is important to figure out the appropriate combination of time and voltage for the desired result from the atmospheric pressure cold plasma treatment. Additionally, temperature and pH can contribute to the transformation of the fats and proteins, changing the viscosity [55]. Relatively steady pH concentrations observed in the present study after plasma treatment support the viscous behavior of milk. However, it is important to measure pH continuously after soaking the absorbent in the sample until FS analysis to provide more accuracy to the statement. Since the changes in viscosity can be attributed to the compositional differences between fats and proteins [51], the current study may foresee fewer chemical changes from plasma application and absorbent use. However, the linear viscoelastic region (LVR) of the control milk was between 0.24 pa and 1.64 pa (Appendix A).

### 3.6. FTIR Analysis

The Fourier-transform infrared spectrum demonstrated a similar pattern between the control and treated samples (Figure 7). The peaks for functional groups of milk compounds in treated samples were also similar to control samples, but at moderately lower absorbance. The fingerprint region of the samples lies within 1400–900 cm^−1^ wavenumbers (Appendix A) and an overlapped spectrum is observed, exhibiting no major change among the samples. It is noteworthy that the fingerprint region can be used to distinguish between compounds as it is unique for a given compound. However, since water was used as a background, there was slight negative absorbance observed, supporting previous research on FTIR analysis of milk [56]. All the peaks detected in the spectrum at different wavenumbers (cm^−1^) is presented in Appendix A.

Fat, protein, lactose, and associated hydrocarbons were predicted in the range of 1400 to 3200 cm^−1^. The peaks at 2923.79 cm^−1^ and 2852.43 cm^−1^ resulted from aliphatic hydrocarbons of the methyl group (CH_2_). Additionally, spectra that feature above 2500 cm^−1^ are expected to be of single bond compounds, i.e., O-H, N-H, and C-H. Due to the lack of a spectral band between 2400 and 2800 cm^−1^ wavenumbers, no compounds were detected, in accordance with previous study [57]. The noisy spectral features from 2300 to 2000 cm^−1^ were most likely caused by diamond absorption with triple bonds [58]. Carbonyl stretching vibration of triglyceride compounds was predicted at 1744.93 cm^−1^ wavenumber [59]. A peak at 1646.84 cm^−1^ was observed near the amide I band of 1660 and 1650 cm^−1^ and is possibly shifted slightly right due to C=O stretching vibrations. Amide I and amide II are the most intense and complex absorption bands of proteins governed by C=O and C-N groups. However, Amide 1 showed a lower absorbance peak than amide II compounds in all the samples. Added to this, both the peak intensities were found to be slightly lower in the treated sample stored for 24 h. The spectral features for carbohydrates and esters were observed between 1000 and 1150 cm^−1^ wavenumbers.

Finding a metal removal technique for foods is preeminent as there have been a very few studies relevant to this. Added to this, heavy metal decontamination from milk is newly introduced in the present study. Utilization of atmospheric pressure cold plasma can be promising upon more extensive research studies. Since foods are more complex in structure than water, the future studies may focus on the foods more. However, the present study could achieve a maximum of 14.89% Cd and 27.37% of Pb removal from milk. Additionally, for water, the maximum reductions for Cd and Pb were 79.85% of and 74.06%, respectively. Cooking methods such as boiling at 100 °C for 15 min and grilling at 180 °C for 20 min could reduce 60% and 61.5% of Pb from shrimps, respectively [60]. Another study reported that pre-cooking processes (washing and rinsing) and cooking together can eliminate 52.6% of Cd and 48.4% of Pb from rice [61]. Compared to these studies, the present study showed lower reduction (%) of heavy metals from food. The use of chelating agents in parallel with cooking methods for metal removal showed promising results too [62]. However, it is noteworthy that most of these removal techniques were utilized for solid foods and require more longer treatment duration than the atmospheric pressure cold plasma.

The use of bio-sorbents in heavy metal removal from wastewater has been reviewed in previous studies [63]. Hence, the present study utilized soybean husk as a bio-sorbent to absorb Pb and Cd from milk. The key mechanism of this process involves the capacity of bio-sorbent and the sorbates’ binding affinity to it. This interaction and affinity were proposedly impacted by the adjacent plasma treatment method, which made the heavy metals available to the sorbent for absorption. Presumably, this correspondence may deteriorate the food quality and the microstructure of it. However, the FTIR spectra and rheological analysis in the current study did not demonstrate substantial difference after the samples were treated with plasma and bio-sorbent absorption adjacently.

## 4. Conclusions

A maximum of 27.37% Pb and 14.89% of Cd were removed from milk samples when the samples were exposed to 50 kV atmospheric pressure cold plasma for 2 min with a 24 h retention time. The low voltage of 20 kV did not have much impact on the metal removal from milk and water. The absorption technique played a major role in the removal, since the 24 h retention time had higher removal (%) than no retention time for milk at 50 kV ACP treatment. However, the milk sample colors were significantly changed from the control sample after the ACP treatment and the absorption. It is possible that the pigments were released from the soybean husk with time, causing the color difference.

The viscosity parameters, measured in terms of storage and loss modulus, did not show a significant difference from the control sample. The storage modulus was dominated by viscous modulus and the complex viscosity did not change substantially with respect to different frequency levels.

Higher voltage and a longer retention period to let the reactive species work on the substance are recommended for future investigation. Further studies are necessary to improve the efficacy of heavy metal removal from liquid food matrices, to identify the suitable time voltage combination, and to distinguish the appropriate mechanism of metal removal from liquid food matrices in order to stay safe from metal toxicity from food consumption.

## Figures and Tables

**Figure 1 biomolecules-12-00913-f001:**
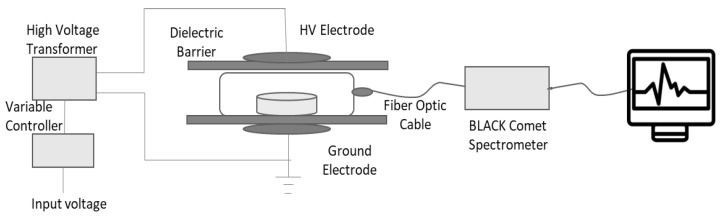
DBD discharge design set up for plasma generation.

**Figure 2 biomolecules-12-00913-f002:**
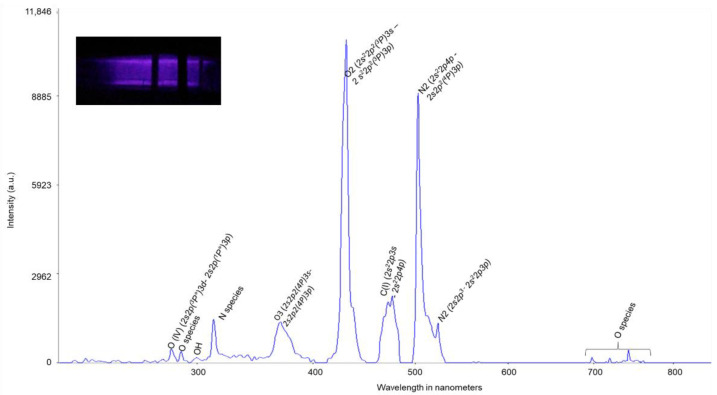
Detection of emitted spectrum in the plasma-treated package (Inset: plasma generation inside the package).

**Figure 3 biomolecules-12-00913-f003:**
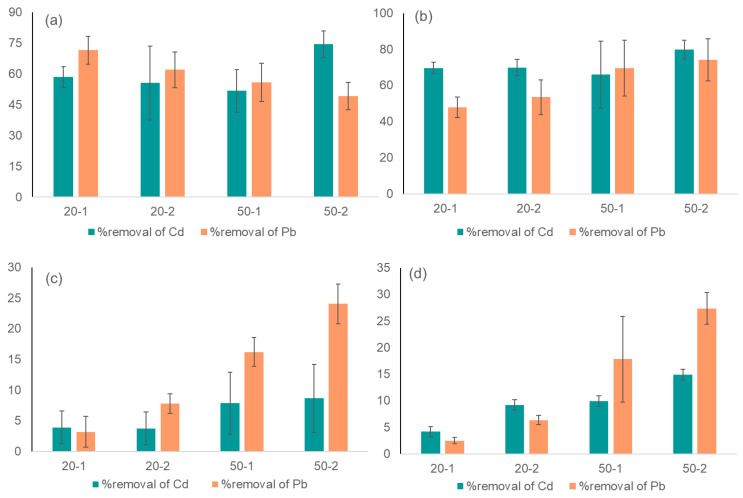
Heavy metal removal (%) after ACP treatment and absorption from the following: (**a**) water for 20 and 50 kV, at 1 and 2 min, with 0 retention time; (**b**) water for 20 and 50 kV, at 1 and 2 min, with 24 h retention time; (**c**) milk for 20 and 50 kV, at 1 and 2 min, with 0 retention time; (**d**) milk for 20 and 50 kV, at 1 and 2 min, with 24 h retention time.

**Figure 4 biomolecules-12-00913-f004:**
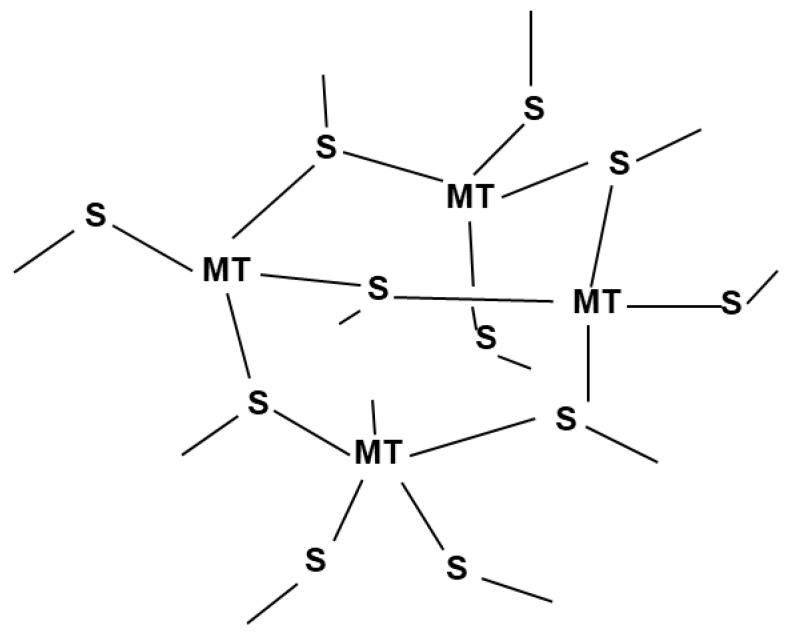
Proposed structure of Metallothionein’s (adopted from Hunt’s study [45]).

**Figure 5 biomolecules-12-00913-f005:**
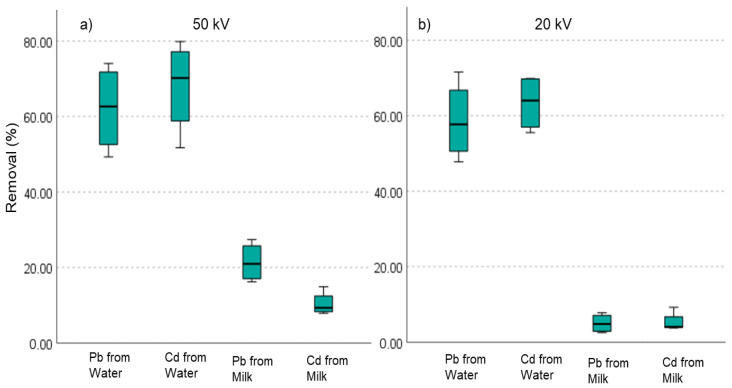
Non-parametric Kruskal–Wallis test for independent samples at (**a**) 20 kV and (**b**) 50 kV (*n* = 6 for each type as no storage time was considered).

**Figure 6 biomolecules-12-00913-f006:**
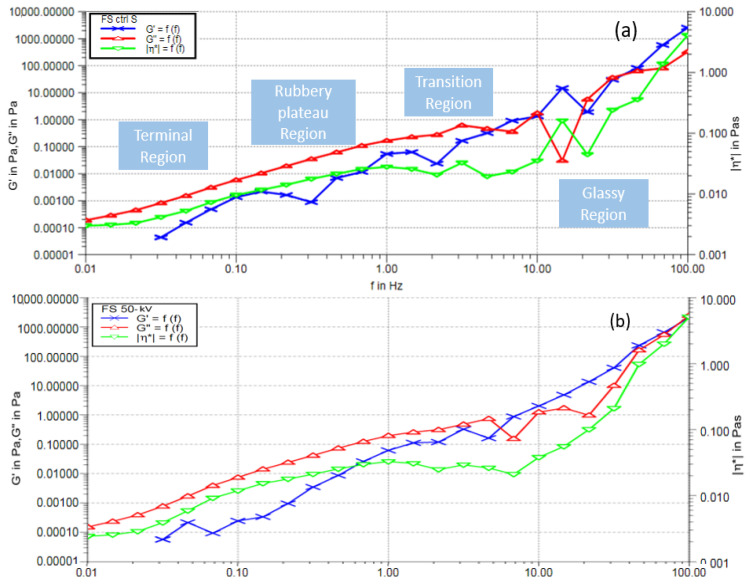
Oscillatory frequency sweep analysis of the milk sample control (**a**) and treated at 50 kV for 2 min and stored for 24 h after treatment/absorbent added (**b**).

**Figure 7 biomolecules-12-00913-f007:**
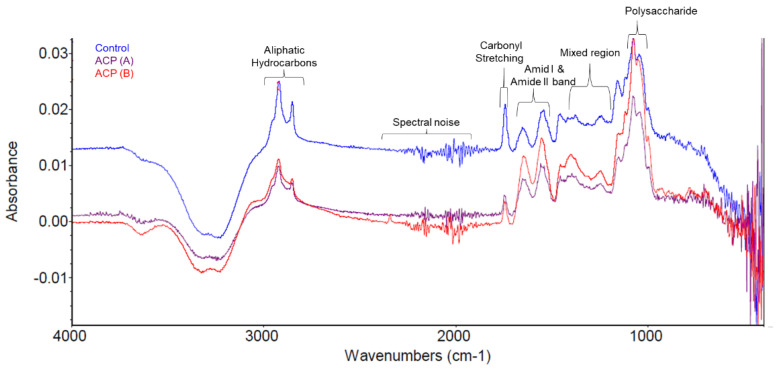
FTIR spectrum of the milk samples. ACP (A) and ACP (B) denotes the sample treated with 50 kV for 2 min with 0- and 24 h storage time, respectively.

**Table 1 biomolecules-12-00913-t001:** Sampling matrix of the atmospheric pressure cold plasma treatment.

Parameters	Treatment Matrix *									
	50 kV				20 kV				Control **	Total
Duration (min)	1 min		2 min		1 min		2min			
Time (h) before Analysis	0	24	0	24	0	24	0	24		
HM in water	0				0				3	3
HM in Milk	0				0				3	3
HM + SB + Water	3				3				6	30
HM + SB + Milk	3				3				6	30

* Abbreviations: HM = heavy metals, SB = soybean husk. ** Controls were without ACP treatment. Pre-treatment HM concentration was also determined for soybean husks. Total Samples: 0.66.

**Table 2 biomolecules-12-00913-t002:** Ozone and pH measurement from the sample at different treatment combinations (Voltage-Time).

	Milk		Water	
kV-min	Ozone (ppm) *	pH *	Ozone (ppm) *	pH *
Control	0	6.6	0	6.3
20-1	10 ± 10	6.5 ± 0.1	10 ± 10	6.2 ± 0.1
20-2	20 ± 10	6.5 ± 0.1	20 ± 10	6.2 ± 0.1
50-1	320 ± 70	6.3 ± 0.1	300 ± 30	6.1 ± 0.1
50-2	500 ± 100	6.3 ± 0.1	520 ± 50	6.1 ± 0.1

* Numbers after ± in the columns denote standard deviation (*n* = 6).

**Table 3 biomolecules-12-00913-t003:** Colorimetric test of the ACP-treated milk samples.

Samples **	L ^*^	a ^*^	b ^*^	ΔE
Control	77.88	−1.91	4.79	-
50-2-24	71.89	−1.67	2.00	6.61
50-1-24	73.25	−1.71	2.58	5.13
20-2-24	71.58	−1.56	2.54	6.70
20-1-24	73.54	−1.60	3.31	4.60
50-2-0	73.85	−1.89	2.21	4.79
50-1-0	73.16	−1.87	1.89	5.54
20-2-0	74.08	−1.85	2.09	4.66
20-1-0	73.86	−1.84	1.92	4.94

** Sample naming explained in Table 1.

## Data Availability

Data supporting the reported results can be found in the Texas A&M University OakTrust Data Repository (Habib et al., 2022).

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
