# Peer review of "Alleviating Heavy Metal Toxicity in Milk and Water through a Synergistic Approach of Absorption Technique and High Voltage Atmospheric Cold Plasma and Probable Rheological Changes"

_biomolecules, 2022, doi:10.3390/biom12070913_

Round 1
Reviewer 1 Report
The manuscript presents effects of N2+O2+CO2 mixture plasma on physiochemical properties of milk and removal of Pb and Cd from milk and water. Plasma was characterized with OES, while tread samples were characterized with a wide range of techniques. The topic is interested and fit with the journal scope. Authors reported good set of experimental work and results. The manuscript aim was clear, the methodology was correct, and the reported results were reasonable. The main defect from my point view, was their carless in writing the manuscript and the art work in the manuscript. In particular section related to the plasma and the OES. The rest of the manuscript is reasonable. I attached some additional remarks about the manuscript.
For instance,
In Abstract, Remove line 12-13 and start with “In this study….
In abstract, line 18, replace 50 KV with “plasma with discharge voltage of 50 kV”
In Abstract Replace “Atmospheric cold plasma” with “Atmospheric Pressure Cold Plasma” , and throughout the manuscript
In abstract, remove line 23-24, and point out to the FTIR, OES, pH and Ozone results
Keywords….need to be revised. Include “Cold Plasma, Spectroscopy, milk, heavy metal removal, rheology”
Please avoid using unpopular abbreviation such as MAP, ACP..
Throughout the manuscript, keep space the value and the unite, for example 60 Hz instead of 60Hz, 4oC ….4oC and also chemical symbols is written carelessly
Page 4, line 128, replace the word “varied” with “range”
Page , line 197….remove the word “output”
Page 5, the sentence “This also resulted in the shifting of the lines through the 209wavelengths, causing limitations of short-lived species identification complexity” is wrong….Please remove it
There is no record for Ozone wavelength at 373.3 and the listed Ref.[29] donot support your claim. Please revise or give other citation. This band is probably SPS N2 Dn=-2
Fig.2, vertical label replaces it with “intensity (a.u.)”
ü In Abstract Replace “Atmospheric cold plasma” with “Atmospheric Pressure Cold Plasma” , and throughout the manuscript
ü Please avoid using unpopular abbreviation such as MAP, ACP..
ü Throughout the manuscript, keep space the value and the unite, for example 60 Hz instead of 60Hz, 4oC ….4oC and also chemical symbols is written carelessly
ü Page 4, line 128, replace the word “varied” with “range”
ü Page , line 197….remove the word “output”
ü Page 5, the sentence “This also resulted in the shifting of the lines through the 209wavelengths, causing limitations of short-lived species identification complexity” is wrong….Please remove it
Author Response
I followed the recommendations and answered the questions. Please see the attached, colored are my answers. Thank you.

Reviewer 2 Report
After careful evaluation of the manuscript, it seems to me that the paper is well written and suitable for Biomolecules after some corrections.
There are some corrections and typos.
1) there is no abbreviation is required in the title (HMs)
2) Suffix of molecules should be written like O2. Rather than “O2”, similarly replace all other molecules present in the text. 4oC should be 4o C, cm-1 should be cm-1
3) Table 1 they shown 30 kV, but tin the text they referred 20 kV, should correct it.
4) Authors can present the table for figure of merit for ACP methods to removal of heavy metal toxicity over the other methods compared with other literatures.
5) Authors stated that “the storage period did not impact the Cd reduction for lower voltages, while for all cases, the storage period impacted the Pb removal”. Mostly authors worked at 20 kV and 50 kV for 1 and 2 minutes at 0-24 hours. Can authors clearly present the which is the optimal parameters for removal of Pb and Cd together.
6) There could be a scope to present the good quality of figures. Figure 2 axis are not visible. Figure 7 can cut after x-axis value 3 and Figure 8 legends cannot be differentiating in black and white print
Author Response
I followed all the instructions and answered the points in color. Please see the attached answer notes.
Thank you.

Reviewer 3 Report
Habib et al. report on the “Alleviating heavy metal (HMs) toxicity in milk and water through a synergistic approach of absorption technique and High Voltage Atmospheric Cold Plasma and probable rheological changes” The content of the work is interesting, but the manuscript cannot be published in the present form due to the following issues:
1. The introduction of the manuscript should be enhanced
2. Why the author has chosen the application of the atmospheric cold plasma technique coupled with the absorption technique in reducing Pb and Cd toxicity from milk and water. The author should involve the novelty and the reliability of the system proposed.
3. Why the Y-Axis has not been shown the Figure 3. It should be simple.
4. Comparative explanations for the different samples have not been mentioned for Figure 6 as control and treated as 50 kV, also please give the physical significance of the change monitored in the two figures.
5. Explain the physical significance of the different curves in Figure 7
6. Figure 8 is blurred please change it with the clear one and also put the first figure in the main manuscript and the other one in the supporting information. Also, label the peaks.
7. Please precisely mention the outcomes regarding work presented in brief as a state of art before the conclusion.
8. Grammar and many typological errors such as subscripts, superscripts, etc. are present in the present form of the manuscript. So need an extensive rectification
Author Response
I followed all the instructions and answered them in color. Please see the attached note.
Thank you.

Round 2
Reviewer 3 Report
As all the comments have been resolved by the author throughout the manuscript. Therefore revised manuscript is acceptable for publication.